# *Hirudo verbana* Microbiota Dynamics: A Key Factor in Hirudotherapy-Related Infections?

**DOI:** 10.3390/microorganisms13040918

**Published:** 2025-04-16

**Authors:** Djursun Karasartova, Gonul Arslan-Akveran, Sabiha Sensoz, Kosta Y. Mumcuoglu, Aysegul Taylan-Ozkan

**Affiliations:** 1Department of Medical Microbiology, Faculty of Medicine, Hitit University, 19030 Corum, Turkey; 2Department of Food Processing, Alaca Avni Celik Vocational School, Hitit University, 19040 Corum, Turkey; gonul.aarslan@gmail.com; 3Department of Nutrition and Dietetics, Faculty of Health Science, Hitit University, 19030 Corum, Turkey; sabiha.sensozz@gmail.com; 4Parasitology Unit, Department of Microbiology and Molecular Genetics, The Kuvin Center for the Study of Infectious and Tropical Diseases, Hadassah Medical School, The Hebrew University, Jerusalem 91120, Israel; kostasm@ekmd.huji.ac.il; 5Department of Medical Microbiology, Faculty of Medicine, International Cyprus University, Nicosia 99258, Cyprus; aysegultaylanozkan@gmail.com

**Keywords:** *Hirudo verbana*, microbiota, hirudotherapy-related infections

## Abstract

The gastrointestinal microbiota of medicinal leeches is particularly interesting due to their blood-feeding habits, increasing medical use, and risk of pathogen transmission. Three groups of *Hirudo verbana* were used to study the leech microbiota: farmed leeches fasting for a long time, farmed leeches recently fed with bovine blood, and wild specimens fed with amphibian blood. The microbiota of the leeches’ mouth, pharynx, crop, and intestine was analyzed. Metasequencing analyses were performed using amplification of the 16S rRNA V3-V4 region on a NovaSeq Illumina platform. The relative abundance of bacterial microbiota included environmental bacteria from the families Rhizobiaceae, Comamonadaceae, Sphingobacteriaceae, Phreatobacteraceae, Myxococcaceae, Chitinophagaceae, Rhodospirillaceae, and Bdellovibrionaceae, as well as symbiotic/probiotic bacteria such as *Mucinivorans*, *Aeromonas*, *Vagococcus*, Lactobacillales, and *Morganella*. Significant differences were found in the different regions of the digestive system among the three groups of leeches, and environmental bacteria were present in all groups to varying degrees. A negative correlation was found between the dominant environmental and the symbiotic/probiotic bacteria. In contrast, a positive correlation was found between environmental and symbiotic/probiotic bacteria, indicating their association with host factors. Microbiota diversity, abundance, and bacterial correlations may be influenced by factors such as the leech’s fasting state, blood meal source, and environmental conditions. The identified opportunistic pathogens, such as *Rickettsia*, *Anaplasma*, and *Treponema*, identified for the first time in *H. verbana*, should be taken into consideration when using this leech in hirudotherapy. Our results show that extensive screening for opportunistic and pathogenic agents should be performed on leeches intended for medical use. Long-fasting leeches and leeches cultured in specialized farms are recommended for hirudotherapy.

## 1. Introduction

Hirudotherapy, practiced for thousands of years, is now officially recognized as a traditional therapy. Currently, hirudotherapy is used worldwide to treat a wide range of acute and chronic diseases [1,2]. In plastic and reconstructive surgery, leeches are an invaluable tool for treating tissue defects, replanting body parts such as fingers, ears, and noses, and relieving venous congestion in skin flaps [3,4]. The Mediterranean medicinal leech (*Hirudo verbana*) is the most widely used and commercially available for hirudotherapy. This species is native to the Mediterranean, Balkan, Levantine, Russian, Armenian, and Turkish regions and plays a central role in modern therapeutic practices [5].

Leeches are a class of segmented invertebrates that parasitize both vertebrates and invertebrates [6,7]. Medical leeches have well developed suckers at their anterior and posterior ends, and their body consists of approximately 34 segments. Their digestive tract consists of the mouth, which is located in the center of the anterior sucker and houses the three muscular jaws, the pharynx, the crop, and the intestine [7]. The crop is the largest compartment of the digestive tract and is responsible for storing digested blood meals and removing water and salt to form a highly viscous intraluminal fluid (ILF). The ILF can remain in the crop for an extended period before gradually entering the intestine, where it is digested [8].

Previous studies on the microbiota of *H. verbana* using both culture-dependent and culture-independent methods have identified a microbiome dominated by *Aeromonas* spp. and *Mucinivorans hirudinis* [9,10,11,12,13]. Metagenomic analyses provide valuable insights into the potential metabolic functions and symbiotic mechanisms between medicinal leeches and their bacterial communities [8]. Among these, *Aeromonas* spp. play a pivotal role by producing hemolysin, which lyses erythrocytes and promotes a symbiotic relationship between the leech and its host [8,14]. Additionally, *M. hirudinis* ferments various sugars, producing alcohols, acetic acid, propionic acid, and succinic acid, which may influence the leech’s gut symbiosis [15]. The leech gut microbiota plays a critical role in several physiological and metabolic processes, including energy production, carbohydrate metabolism, amino acid and lipid synthesis, and inorganic ion transport. It also contributes to the biosynthesis of secondary metabolites and essential vitamins [8]. Resident symbiotic bacteria in the gut produce antimicrobial peptides that inhibit the growth of competing bacteria, including potential pathogens. This activity indirectly supports the health and well-being of the leech host. In addition, the role of these symbiotic bacteria in protecting the gut from pathogen colonization has been demonstrated in medicinal leeches [12]. Several low-abundance taxa, such as *Phreatobacter*, *Taibaiella*, *Fluviicola*, *Aquabacterium*, *Burkholderia*, *Hydrogenophaga*, *Wolinella*, and *Chitinophagia*, have been detected in the crop and intestine [13]. In addition, symbionts such as Proteobacteria, Fusobacteria, Firmicutes, and Bacteroidetes have also been identified [9]. Taxonomic groups such as *Proteus*, *Clostridium*, *Erysipelothrix*, *Desulfovibrio*, and *Fusobacterium* have been detected in the ILF and intestine of *H. verbana* [11]. Strictly hematophagous leeches, such as the medicinal leech *H. verbana* (Hirudinea: Hirudinidae), represent a naturally simplified and robust model for investigating host–microbe symbioses within the digestive tract due to their limited and specialized microbiota [8,9,11,13].

While adhering to established standards is crucial, the risk of infection during treatment remains insufficiently explored. While leeches offer significant therapeutic benefits, their clinical use is sometimes associated with wound infections and serious complications such as meningitis, bacteremia, and sepsis. These adverse outcomes are thought to be due to the symbiotic bacteria in their digestive tract. Reported rates of leech-associated infections in the literature vary widely, ranging from 2.4% to 36.2%. The majority of these infections are caused by *Aeromonas* spp. However, other Gram-negative bacteria such as *Pseudomonas* spp., *Proteus* spp., *Klebsiella* spp., *Serratia* spp., *Enterobacter* spp., and *Morganella* spp., as well as Gram-positive bacteria such as *Staphylococcus* spp. and *Enterococcus* spp., have also been implicated [16,17,18].

A comprehensive investigation of the gastrointestinal microbiota of *H. verbana* is crucial, yet extensive studies on this topic remain scarce. Most existing research has primarily focused on analyzing the crop and intestinal microbiota after leech feeding [9,11,13]. However, since medicinal leeches are often used in an unfed state, understanding the microbiota composition across the entire digestive tract—mouth, pharynx, crop, and intestine—under both fasting and blood-fed conditions is essential. The aim of this study is to investigate the dynamics of the microbiota of the entire digestive tract of *H. verbana* depending on environmental conditions, nutritional status, and the host-derived blood source, as well as the identification of opportunistic microorganisms. It is assumed that knowledge of microbiota dynamics and opportunistic pathogens would form the basis for the safety and success of hirudotherapy by preventing hirudotherapy-related infections.

## 2. Materials and Methods

### 2.1. Leeches

A total of 80 *H. verbana* were purchased from a certified leech breeding farm in Istanbul. Of these, 40 leeches had been fed six months earlier (=unfed), while 40 leeches had been fed 7 days earlier with bovine blood (=fed). In addition, 40 leeches were purchased from a breeder in the Thrace region, where unidentified frogs were used as hosts in natural ponds (=wild). The farmed leeches were transferred to a tank of sterile tap water, while the wild leeches were kept in their pond water. All leeches were morphologically identified as *H. verbana* [6,19] and kept in the laboratory for one week before being used for further analysis. In addition, the manufacturers provided certificates confirming that the leeches were produced as *H. verbana*.

### 2.2. Collection of Samples from the Leeches

To anesthetize the leeches without causing them to regurgitate, 70% alcohol was slowly added to the water containing the live leeches, and the concentration gradually increased for about 30 min until the leeches stopped moving [6]. Thereafter, the leeches were thoroughly rinsed with RNase/DNase-free molecular biology-grade water for 30 s. Sterile instruments and solutions were used during the dissection of the leeches, and all procedures were performed in a biosafety level 2 cabinet.

The unfed leeches were positioned ventrally on a sterile board and fixed with needles at both ends. A 2–3 cm longitudinal incision was made along the ventral surface of the leech, beginning approximately 1 cm below the anterior sucker. The mouthparts, located in the center of the anterior sucker and including the three jaws, were removed by dissecting the first 2–3 mm of the apical part of the body. Skin and connective tissue were removed from the area of the pharynx, crop, and intestine. The pharynx is approximately located between the 5th and 8th segments, the crop between the 9th and 19th segments, and the intestine between the 20th and 26th segments [19,20,21,22,23], were removed using separate dissection tools. Tissue samples were taken and homogenized by utilizing a sterile scalpel.

Fed samples were processed as described above. Since the gastrointestinal system of the leeches was filled with the intraluminal fluid (ILF), 100 μL samples were taken with a pipette through a small incision on the ventral surface of the crop and intestine, while the mouthparts together with the three muscular jaws and pharyngeal regions were removed from the leech body. All dissections were performed with a professional 30× magnifier with LED light (AIXPI, Shenzhen, China).

### 2.3. DNA Extraction and Pools

In each group, 40 mouth parts, pharynx, crop, and intestine (homogenized tissue or ILF) were pooled in four vials (Figure 1 and Table 1). DNA extraction was performed using tissue and bacterial DNA purification kits (EURx, Gdansk, Poland) according to the manufacturing protocol under sterile conditions in a class II biological safety cabinet (Thermo Scientific, Langenselbold, Germany). The extracted DNA was analyzed on a 1% agarose gel, and the DNA concentration and purity were measured using a NanoDrop spectrophotometer (Thermo Scientific, Wilmington, DE, USA). Then, the DNA samples were stored at −20 °C until sequencing.

### 2.4. PCR Amplification and Sequencing (Next-Generation Sequencing (NGS))

The DNA was subjected to the initial PCR process, targeting the V3-V4 region of the 16S rRNA with specific primers:

F (5′TCGTCGGCAGCGTCAGATGTGTATAAGAGACAGCCTACGGGNGGCWGCAG) and R-(5′GTCTCGTGGGCTCGGAGATGTGTATAAGAGACAGGACTACHVGGGTATCTAATCC′) [24].

The PCR procedure was performed using KAPA HotStart PCR Mix and DNA samples, with the following cycle implemented: 95 °C for 3 min; 95 °C for 30 s, 55 °C for 30 s, 72 °C for 30 s for the subsequent 25 cycles, and 72 °C for 5 min. The PCR products were subjected to electrophoresis for verification purposes. To remove primer dimers and free primers, two rounds of AMPure XP magnetic beads (Beckman Coulter, Brea, CA, USA) were used during the clean-up phase of the PCR product purification process. The purified products were then subjected to index PCR using the Nextera XT Index Kit (Illumina, San Diego, CA, USA). The following reaction was performed using DNA, Index 1 and 2 primers, KAPA HotStart Ready Mix, and PCR-grade water. Thereafter, they were subjected to a temperature of 95 °C for three min, followed by eight cycles of 95 °C for 30 s, 55 °C for 30 s, 72 °C for 30 s, and a final step of 72 °C for 5 min. Subsequent purification was conducted using AMPure XP beads. PCR products were sequenced in a Novaseq System (Macrogen, London, UK) for next-generation massive sequencing.

Three control samples were prepared: a. the reagent control, which was prepared by performing the DNA extraction procedure using the same reagents but without including the leech sample; b. the negative PCR controls, which were prepared by performing PCR amplification using molecular biology grade water; and c. the bovine blood. The two negative controls and the bovine blood did not produce any bands in the two-step PCR described above, indicating the absence of external bacterial contamination.

### 2.5. Bioinformatics Analyses

The Illumina Novaseq system USA (NovaSeq Control Software v1.8.0, Illumina, San Diego, CA, USA) was used to analyze the readout images. Bases were determined using Real-Time Analysis v1.18 software, while DNA sequences were examined using the Quantitative Insights into Microbial Ecology II (QIIMEII-amplicon-2024.10) bioinformatics software. The files were converted to FASTA format using the program bcl2fastq (v1.8.4), and all chimeric sequences were discarded. Non-specific adaptor sequences were removed from the reads using the programs Scythe (v0.994 BETA) and Sickle. The quality of the raw data (FASTA format) was assessed using the FastQC/v0.11.8 tool [25]. The microbial 16S data were further filtered and trimmed according to the error rate, after which the amplicon sequence variants (ASVs) were identified using the denoising algorithm as implemented in DADA2 [26]. Taxonomic analysis of the ASVs was performed using QIIME2 (v2022.2) software [27]. Taxonomy was assigned using the SILVA 138 16S rRNA gene reference database [28,29]. Figure 2 shows the results of the alpha rarefaction analysis, which assesses the adequacy of the sequencing depth for the samples. The parallel lines in the alpha diversity plot on the right indicate that the sequencing depth is sufficient for further analysis.

### 2.6. Statistical Analysis

Chi-square tests were applied to evaluate significant differences in the relative abundances of major bacterial phyla across the leech groups.

Alpha diversity was calculated using QIIME2 with metrics such as Shannon and Simpson indices; to assess species richness and evenness within individual samples and compare alpha diversity among the different groups, a one-way ANOVA test was conducted. Post hoc pairwise comparisons were performed using Tukey’s Honest Significant Difference (HSD) test following the ANOVA to identify significant differences between specific groups. To compare the abundance of environmental and symbiotic/probiotic bacteria among different groups, a Kruskal–Wallis rank sum test was performed due to the non-normal distribution of the data. Following a significant Kruskal–Wallis result (*p* < 0.05), pairwise comparisons were conducted using the Wilcoxon rank sum test with Bonferroni correction to adjust for multiple comparisons. The significance level of *p* < 0.05 was considered statistically significant. The results of the pairwise Wilcoxon tests were used to classify groups into statistically distinct subsets, which are indicated by letters above the boxplots.

Beta diversity was analyzed to investigate differences in microbial community composition between groups. Distance matrices were generated using metrics such as Bray–Curtis and weighted and unweighted UniFrac within QIIME2. Principal coordinate analysis (PCoA) and Non-Metric Multidimensional Scaling (NMDS) were used to visualize differences in community structure, generated using the QIIME diversity core-metrics-phyloseq package. The statistical significance of group differences in beta diversity was assessed using the PERMANOVA test, performed with the QIIME diversity beta-group-significance plugin, with permutations. Rarefaction analysis was conducted to determine the adequacy of sequencing depth and to ensure comparability between samples. The generation of rarefaction curves was facilitated by the QIIMEII diversity alpha-rarefaction plugin in QIIME2, and the sequencing data were rarefied to a uniform depth prior to further analysis. Spearman’s rank correlation coefficient (ρ) was calculated to assess monotonic relationships between the variables. The correlation matrix was computed using Python’s scipy. stats. spearmanr function, and statistical significance was set at *p* < 0.05. To visualize the correlation results, a heatmap was generated, where the color gradient represents the strength and direction of the correlation coefficient (*ρ*), ranging from −1 (strong negative correlation) to +1 (strong positive correlation). Significant correlations were highlighted using different color thresholds. The heatmap was generated using Python’s Seaborn library (version 0.12.2, Python 3.10). A clustered heatmap was generated based on the relative abundance (%) of bacterial taxa. Hierarchical clustering was performed using Euclidean distance and average linkage.

## 3. Results

### 3.1. Microbiota

A total of 120 sections of the *H. verbana* digestive system (mouth, pharynx, crop, and intestine) were pooled into 12 samples and used for bacterial microbiota sequencing. Sequences were assembled at an average count of 122,006. After taxonomic classification, 93,305 bacterial sequences were obtained (Appendix A). A total of 5878 amplicon sequence variants (ASVs) were identified with 33 phyla, 72 classes, 168 orders, 313 families, and 553 genera. Overall, 96.3% of the identified microorganisms were bacteria, 1.8% were eukaryotes, and 1.9% were unclassified.

### 3.2. Relative Abundance of Bacterial Taxa at Phylum Level

Of the 33 phyla identified, 31 (93%) were taxonomically classified, while the remainder were classified as “other”. Bacterial microbiota from 12 pools of *H. verbana* showed that Proteobacteria was the dominant phylum with a relative percent abundance (RPA) of 51%, followed by Bacteroidota (31%), Firmicutes (6%), Myxococcota (3%), Bdellovibrionota (1%), and Spirochaetota (1%) (Figure 3). Proteobacteria and Bacteroidetes were present at high densities throughout the digestive system of all leech groups. Myxococcota species were the most abundant bacteria (35%) in the mouth of farmed fasting leeches. Firmicutes were identified in the mouth of farmed fasting leeches (6%), in all parts of the digestive system of farmed newly fed leeches (9–17%), and in the pharynx (8%) and intestine (4%) of field-collected leeches, while Bdellovibrionota (1–8%) and Spirochaetota (1–4%) were found predominantly in field-collected leeches (Figure 3). Chi-square test results indicate that there are no significant differences in Proteobacteria and Bacteroidota abundances among leech groups. However, Myxococcota was significantly more abundant in the farmed unfed group compared to both farmed fed and wild fed groups (*p* < 0.001). Firmicutes showed a significantly higher abundance in the farmed fed group compared to both farmed unfed (*p* < 0.001) and wild fed (*p* = 0.01) groups.

### 3.3. Relative Abundance (%) of Bacteria at Family and Genus Level

The relative abundance of the bacterial microbiota of farmed, both fed and unfed, and wild *H. verbana* at the family and genus level were as follows: Rhizobiaceae (18%), Comamonadaceae (15%), *Mucinivorans* (14%), *Nubsella* (12%), *Phreatobacter* (3%), Myxococcaceae (3%), *Aeromonas* (7%), Chitinophagaceae (2%), Rhodospirillaceae (2%), *Bdellovibrio* (1%), *Vagococcus* (2%), Lactobacillales (2%), *Rickettsia* (1%), *Morganella* (1%), *Haoranjiania* (1%), and other bacteria (12%) (Figure 4). Rhizobiaceae, Comamonadaceae, *Nubsella*, and Chitinophagaceae, which are known environmental bacteria from water and soil habitats, were generally more abundant in the digestive tract of fasting leeches (mean 20%, 22%, 17%, and 3%, respectively) and field-collected leeches (mean 27%, 16%, 14%, and 2%, respectively). In contrast, they were less abundant in farmed leeches (mean 5%, 9%, 6%, and 0.25%, respectively). Rhizobiaceae, Comamonadaceae, and *Nubsella* were the dominant groups in the mouths of field-collected leeches at 46%, 17%, and 16%, respectively. *Phreatobacter* was detected in an average of 5% of fasting leeches, 3% of recently fed leeches, and 3% of field-collected leeches. Rhodospirillaceae was found in 7% and 11% of the intestines of field-collected and farmed fasting leeches, respectively. Myxococcaceae dominated (35%) the mouth of farmed fasting leeches. *Bdellovibrio* was detected only in the pharynx (1%), crop (4%), and intestine (7%) of field-collected leeches. The genus *Mucinivorans* was more abundant in farmed fed leeches (mean 24%) than in farmed unfed leeches (mean 9%) and wild-fed leeches (mean 8%). *Morganella* were present at 2% in farmed fed *H. verbana* specimens, while *Aeromonas* species were more dominant, with relative abundances of 17%, 17%, 24%, and 20% in the mouth, pharynx, crop, and intestine, respectively. *Aeromonas* spp. were detected in very low quantities (0.02–0.4%) in fasting and wild leeches. In comparison, *Vagococcus* and Lactobacillales were detected only in farmed fed leeches (mouth 1% and 3%, pharynx 2% and 6%, crop 6% and 9%, and intestine 5%, respectively). Bacillaceae were found in the mouth (4%) of the same leech. *Haoranjiania* sp. was detected in the mouth of fed (1%), unfed (5%), and wild (3%) *H. verbana* specimens, while *Lactobacillus* was found in the mouthparts (3%) of unfed leeches (Figure 4).

Overall, a significant difference in alpha diversity was detected among the fed, unfed, and wild groups of leeches (*p* = 0.004) (Figure 5). However, no significant differences were found in the Shannon (*p* = 0.29) and Simpson (*p* = 0.11) indices. Post hoc Tukey’s HSD test revealed that the farmed unfed group exhibited significantly lower alpha diversity compared to both the farmed fed and wild fed groups (*p* < 0.05). In contrast, no significant difference was observed between the farmed fed and wild fed groups (*p* > 0.05).

No significant differences in microbial diversity were found between the oral, pharyngeal, crop, and intestinal regions of individual leeches (fed *p* = 0.99; unfed *p* = 0.55; wild *p* = 0.24). Post hoc Tukey tests revealed specific differences among digestive regions, indicating that despite the overall non-significant *p*-values, certain microbial diversity variations exist within individual leech groups (Table 2). The results indicate that farmed fed leeches exhibited the highest microbial diversity, with significantly elevated Shannon and Simpson indices across multiple anatomical regions, particularly in the mouth and pharynx. This suggests that feeding under controlled conditions promotes a stable and enriched microbial community. In contrast, farmed unfed leeches displayed the lowest microbial diversity, implying that prolonged starvation may reduce microbial richness. Microbial diversity differed significantly between the mouth and crop in the farmed unfed group. In the wild fed group, the mouth exhibited the lowest microbial diversity and was statistically distinct from the crop and intestine. These findings highlight the significant influence of feeding status and environmental factors on the digestive system’s microbiota of *H. verbana*, emphasizing the role of diet in shaping microbial diversity and stability (Table 2).

The results of the PERMANOVA analysis of beta diversity showed that there were statistically significant differences between the mouth, pharynx, crop, and intestine (*p* = 0.001) between leech groups. This high (76.5%) (*R*^2^ = 0.76522) variance and cluster separation between the leech body regions were also confirmed by the PCoA (Bray–Curtis) and NMDS (Bray–Curtis) plots (Figure 6 and Appendix A, respectively). These results suggest that environmental factors such as starvation and food sources have a significant impact on bacterial diversity.

The Kruskal–Wallis test revealed a statistically significant difference between the leech groups in the abundance of the most common environmental bacteria (Rhizobiaceae, Comamonadaceae, *Nubsella*, and Chitinophagaceae) (χ^2^ = 12.185, df = 2, *p* = 0.00226) (Figure 7). Furthermore, the same test revealed a statistically significant difference in the abundance of the most common symbiotic/probiotic bacteria (*Aeromonas*, *Mucinivorans*, *Morganella*, *Vagococcus* and Lactobacillales) between the groups (χ^2^ = 31.212, df = 2, *p* < 0.001) (Figure 8).

Environmental bacteria were abundant in farmed unfed and wild fed leeches, while symbiotic/probiotic bacteria were low. In contrast, farmed fed leeches showed low environmental but high symbiotic/probiotic bacterial abundance. There was a negative correlation between symbiotic/probiotic bacteria and environmental bacteria (Figure 9). The average Spearman correlation coefficient was −0.683 and *p*-value = 0.00091. A positive correlation was found between the symbionts *Aeromonas* and *Mucinivorans* (+0.753, *p* = 0.01); *Mucinivorans* and *Vagococcus* (+0.755, *p* = 0.05); *Mucinivorans* and Lactobacillales (+0.783, *p* = 0.005); *Aeromonas* and *Vagococcus* (0.998, *p* = 0.001); *Aeromonas* and Lactobacillales (+0.987, *p* = 0.001); *Vagococcus* and Lactobacillales (+0.993, *p* = 0.001); *Morganella* and *Aeromonas* (+0.977, *p* = 0.001); *Morganella* and *Mucinivorans* (+0.790, *p* = 0.005); *Morganella* and *Vagococcus* (+0.969, *p* = 0.001); *Morganella* and Lactobacillales (+0.968, *p* = 0.001). A positive correlation was also found between abundant environmental bacteria: Rhizobiaceae and Comamonadaceae (+0.650, *p* = 0.05); Rhizobiaceae and *Nubsella* (+760, *p* = 0.01); *Nubsella* and Comamonadaceae (+1.00, *p* = 0.001); Chitinophagaceae and Rhizobiaceae (+0.596, *p* = 0.005); Chitinophagaceae and *Nubsella* (+0.601, *p* = 0.05); Chitinophagaceae and Comamonadaceae (+0.350, *p* = 0.5). Correspondingly, a positive correlation was found within the symbiotic/probiotic bacteria and also within the abundant environmental bacteria (Figure 9).

Correlations between host factors and bacterial abundance were also analyzed, as illustrated in Figure 10. Overall, with the results mentioned above, wild and unfed farmed leeches showed positive correlations with environmental bacteria and negative correlations with probiotic/symbiotic bacteria. İn addition, wild leeches were positively associated with *Bdellovibrio*, *Rickettsia*, and *Treponema*. In contrast, fed farmed leeches showed positive correlations with core probiotic/symbiotic bacteria, as well as *Wolinella*, *Treponema*, and Enterobacterales.

### 3.4. Opportunistic Pathogenic Bacteria That May Cause Infection During Hirudotherapy

Bacteria such as Aeromonas, Morganella, Pseudomonas, Acinetobacter, Staphylococcus, Fuzobacterium, Porphyromonas, Clostridia, Bacteroides, Anaplasma, Rickettsia, and Treponema were identified in various quantities (Table 3).

## 4. Discussion

### 4.1. H. verbana Microbiota Diversity

In the present study, the bacterial microbiota of the different digestive tract regions (mouth, pharynx, crop, and intestine) of *H. verbana* was examined, and the presence of opportunistic pathogens was investigated. Previous studies have shown that medicinal leeches have a relatively limited bacterial microbiota compared to other animals [8,30] and that the gut of this hematophagous annelid is dominated (>95%) by two species of gammaproteobacteria, *Aeromonas* and *Mucinivorans*, whose spatial and temporal population dynamics have been described during blood digestion [9,31,32]. In addition, *Morganella morganii* was identified alongside the two symbionts mentioned above, albeit in lower abundance [9,33,34]. The analysis of crop and gut samples of fed farmed *H. verbana* revealed the presence of bacteria such as *Proteus*, *Clostridium*, *Eryspelothrix*, *Desulfovibrio*, and *Fusobacterium* [11], while in wild *H. verbana* samples, *Phreatobacter*, *Taibaiella*, *Fluviicola*, *Aquabacterium*, *Burkholderia*, *Hydrogenophaga*, *Wolinella*, and *Chitinophagia* were identified [13]. In our study, the most common families and species were Rhizobiaceae (18%), Comamonadaceae (15%), *Mucinivorans* (14%), and *Nubsella* (12%). Overall, these results indicate that environmental bacteria are present in significantly higher abundance than symbiotic and probiotic bacteria. These results contrast with previous studies on *H. verbana* leeches. Possible explanations for this discrepancy may be found in geographic and environmental factors, the type of food source, the feeding conditions of the leeches, and the inclusion of all parts of the leech digestive system. The environmental bacteria detected in the study are predominantly found in aquatic and soil ecosystems [35,36,37,38,39,40]. The most abundant bacteria, Rhizobiaceae, are characterized by their ability to establish symbiotic relationships with host plants and facilitate the process of biological nitrogen fixation [37]. The other abundant bacteria are Comamonadaceae, which are characterized by their ability to utilize diverse metabolic pathways and energy production mechanisms, thereby playing a pivotal role in the cycling of carbon, nitrogen, iron, and other elements within ecosystems; they are also aerobic organotrophs, photoautotrophic and photoheterotrophic bacteria, and fermentative bacteria [36]. *Nubsella* plays a key role in the organic carbon cycle within its environmental niche. It has the ability to facilitate the degradation of environmental organic matter through the production of various hydrolytic enzymes [35]. The presence of these environmental bacteria indicates that the bacterial community in the digestive tract of the leech is shaped by the freshwater environment [13]. Alternatively, these bacteria may function as symbiotic microorganisms within the digestive tract of *H. verbana*. Another study identified members of the Rhizobacteriaceae family in the bladder of the North American medicinal leech *Macrobdella decora*. In addition, *Nubsella* and Comamonadaceae were found to be the core microbiome in the bladder of *H. verbana* [32].

### 4.2. Aeromonas and Mucinivorans Abundance

The results of the present study showed that farmed fed leeches were positive for *Aeromonas* and *Mucinovarans.* This finding is consistent with the conclusions of previous studies in which bovine blood was the primary source of leech feeding [9,11,32,41]. Notably, in wild fed leeches, *Aeromonas* was in trace amounts, while *Mucinivorans* dominated their digestive tract, though less abundantly than in farmed fed *H. verbana.* A previous study reported abundant *Aeromonas* and *Mucinivorans* in wild *H. verbana*, though the host species remained unidentified [13]. The presence of *Aeromonas* was only detected at negligible levels in the mouth of unfed leeches, most likely due to the absence of blood in the digestive tract. The results of this study suggest that the hungrier the leeches used for medicinal purposes, the lower the concentration of *Aeromonas* bacteria. It is important to note that *Aeromonas* is the most common cause of infection during hirudotherapy [42], underscoring the importance of controlled rearing conditions for medicinal leeches. *Aeromonas* load was low in fasting animals, and its abundance was highest after blood feeding [12,42]. Porcine blood-fed *Hirudo nipponica* showed increased *Mucinivorans* and *Aeromonas* levels compared to pre-feeding [43]. Following feeding, *Aeromonas veronii* and *Mucinovorans hirudinis* rapidly proliferate within three days; *M. hirudinis* peaks at 7 days and then gradually declines. *A. veronii* decreases within day 14, while *M. hirudinis* declines more slowly [20,44]. In the present study, *Mucinovarans* were less abundant in farmed leeches after six months of starvation than in *H. verbana* farmed fed leeches.

### 4.3. Beta and Alpha Diversity

Beta and alpha diversity analyses revealed statistically significant differences between digestive regions (mouth, pharynx, crop, and intestine) across the three leech groups. These variations are likely influenced by nutritional status, blood meal source, and environmental factors. However, overall microbial diversity differences within individual leech groups were not statistically significant. This may be related to their anatomy, consisting of a single, unbranched digestive tube. This compact structure enables efficient food passage, and regurgitation behavior is also observed during digestion [45,46]. Post hoc analyses revealed region-specific variations. Farmed fed *H. verbana* exhibited the highest microbial diversity, particularly in the mouth and pharynx. In contrast, microbial diversity was lowest in farmed unfed animals. The mouth showed distinct microbial profiles compared to other regions in both unfed and wild fed leeches. Observations indicate that microbial diversity varies within individual leech groups, with controlled feeding enhancing diversity and stability, whereas starvation and environmental exposure lead to reduced diversity and region-specific alterations. These results highlight the critical role of diet in shaping gut microbial communities in *H. verbana.*

In farmed unfed groups, *Myxococcaceae* dominated, reaching 35% abundance in the mouthparts. These bacteria are found in soil, freshwater, and marine environments [40]. In general, environmental bacteria were present throughout the digestive system of unfed animals. Furthermore, Rhizobiaceae, Comamonadaceae, and *Nubsella* were dominant in the mouth of wild leeches. In particular, *Rickettsia*, *Bdellovibrio*, Synergistaceae, Selenomonadales, and Desulfovibrioaceae were identified mainly in the crop and intestine of wild specimens. These findings suggest that microbiota dynamics are shaped by environmental conditions and dietary inputs. *Rickettsia* spp. are common intracellular pathogens of animals, humans, plants, and also facultative and sporadic symbionts of invertebrates [47]. *Bdellovibrio* species are known as predatory bacteria that feed predominantly on Gram-negative/positive organisms [48]. In addition, the safety of *Bellovibrio* has been demonstrated for mammals and sturgeons, and it has a wide range of prey and significant bacteriolytic activity against *Aeromonas hydrophila* [49]. This might explain why *Aeromonas*, the key bacterium responsible for blood digestion, is present in negligible quantities in wild leeches. On the other hand, this feature suggests that it may be a solution against *Aeromonas* spp., which are the most common causes of infections in hirudotherapy. Desulfovibrioaceae are found in various habitats, including freshwater, brackish, marine sediments, soil, and animals. The majority of species described are chemoorganoheterotrophs, with some being chemolithoheterotrophs, and all of them oxidize organic substrates to acetate [50]. Our results show the presence of environmental bacteria in all three groups of leeches, suggesting that these bacteria may be part of the permanent flora of *H. verbana*. However, these bacteria can be replaced by symbiotic or probiotic species under certain conditions. While the absence of symbiotics/probiotics in farmed unfed leeches was expected, their scarcity or non-detection in wild fed leeches is noteworthy.

### 4.4. Correlation Dynamics

Abundant environmental bacteria dominated in farmed unfed and wild fed leeches, while symbiotic/probiotic taxa were scarce. In contrast, farmed fed leeches showed reduced environmental bacteria but a higher abundance of symbiotic/probiotic bacteria. A clear negative correlation was observed between environmental and symbiotic/probiotic bacteria. After feeding *H. verbana* with bovine blood, the proliferation of *Aeromonas*, *Morganella*, and *Vagococcus* was observed from day 7 to day 90 [9]. *Mucinivorans* can ferment glucose and lactose, similar to *Vagococcus* and *Lactobacillales* [51]. It also showed a synergistic relationship with *Aeromonas*, both abundant in farmed fed leeches. *Aeromonas* was found only in trace amounts in farmed unfed and wild leeches, matching the low levels of *Mucinivorans*. This suggests a cooperative relationship among *Mucinivorans*, *Aeromonas*, *Morganella*, *Vagococcus*, and *Lactobacillales*, supporting each other’s growth. Positive correlations were observed not only among symbiotic/probiotic bacteria but also among dominant environmental taxa. The observed positive and negative correlations among bacterial communities were found to be influenced by host factors, underscoring the pivotal role of the host environment in shaping microbial interactions. The gut microbiota has been shown to play a central role in shaping the intestinal immune response in mammals [52,53]. *Aeromonas* produce antibiotic substances that provide protection against invasive bacteria and contribute to the distinctive simplicity of the leech gut microbiota [12]. Resident commensal microorganisms not only synthesize antimicrobial substances but also compete for nutrient resources, space, and cellular receptors in the gastrointestinal system, thus implementing their anticompetitive function in relation to other bacteria, including pathogens [54]. This study suggests that in farmed leeches, symbionts such as *Mucinivorans*, *Aeromonas*, *Morganella*, *Vagococcus*, and *Lactobacillales* interact to suppress environmental bacteria. Conversely, environmental bacteria may inhibit symbiotic taxa in wild fed and unfed leeches through similar mechanisms. We hypothesize that bacterial correlations are influenced by the host’s nutritional status, blood-derived nutrients, and ecological conditions. Understanding these interactions may offer strategies to suppress pathogens like *Aeromonas* and *Morganella*, which are associated with post-hirudotherapy infections. It is important to note that the colonization of bacteria in the digestive system of the leech is also influenced by the complement system of ingested vertebrate blood, which remains active in the leech for a while and can effectively kill sensitive bacteria. The animal’s microbiota can be shaped by this factor, emphasizing the importance of the blood source on which the leech feeds [55].

### 4.5. Infection Risk Assessment of Hirudotherapy

In our study, the farmed fed group appeared to be the most at risk in terms of potential pathogens, showing notable abundances of *Aeromonas*, *Morganella*, *Bacillaceae*, and *Treponema*, along with lower levels of *Bacteroides*, *Pseudomonas*, *Acinetobacter*, *Fusobacterium*, and *Anaplasma*. The wild fed group was characterized by bacteria such as *Rickettsia*, *Treponema*, and *Clostridia* in notable amounts, *Porphyromonas*, and *Anaplasma* at lower levels. In addition, both the farmed fed and wild fed groups harbored other opportunistic bacteria, albeit in trace amounts, highlighting the persistent risk of low-level pathogen presence under rearing or environmental conditions. The farmed unfed group exhibited the lowest diversity of pathogenic taxa, suggesting that starvation may contribute to microbial suppression. Additionally, rearing leeches under hygienic and controlled farm conditions is essential to ensure microbiological safety. In the present study, aerobic bacteria such as *Rickettsia*, *Anaplasma*, and *Treponema* were detected for the first time in *H. verbena*. It has been reported that rickettsial disease occurred after being bitten by a terrestrial leech (*Haemadipsida* spp.) [56]. In another study, *Rickettsia* was detected in 96% of glossiphoniid leeches [57]. *Treponema* species are commensals of the human and bovine gastrointestinal microbiome [58,59]. These newly identified pathogens can be harbored by *H. verbana*, suggesting that they may also cause hirudotherapy-related infections.

## 5. Conclusions

This study highlights the significant impact of feeding status, blood source, and environmental exposure on the digestive microbiota of *H. verbana*. Correlations among dominant microbial taxa and their associations with host-related factors reveal microbial dynamics influenced by physiological and external conditions. Detecting opportunistic pathogens such as *Rickettsia*, *Anaplasma*, and *Treponema* underscores the importance of assessing the safety of hirudotherapy. These findings emphasize the need to optimize fasting protocols, blood meal selection, and environmental management to reduce infection risks and enhance therapeutic outcomes; however, further comprehensive studies are required to validate these results.

## Figures and Tables

**Figure 1 microorganisms-13-00918-f001:**
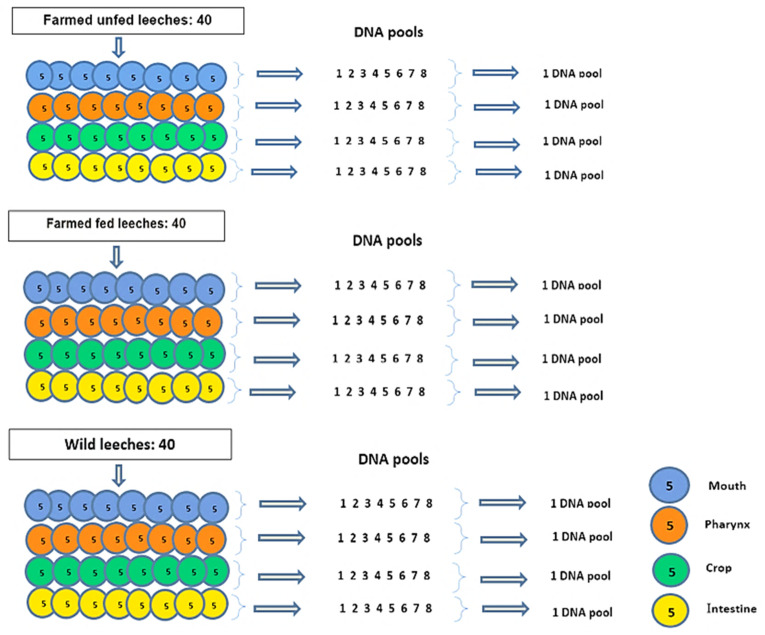
A total of 120 leeches were used: 40 farmed unfed, 40 farmed fed, and 40 wild leeches. In each group, the following procedure was applied: Five leeches were dissected, and their mouth, pharynx, crop, and intestine tissues were pooled by region. DNA was extracted from each pooled tissue sample. This procedure was repeated eight times per group, resulting in eight DNA pools per tissue and per group. Finally, for each body part (mouth, pharynx, crop, intestine) within each group, the eight DNA pools were combined into a single pool. This yielded a total of 12 final DNA pools.

**Figure 2 microorganisms-13-00918-f002:**
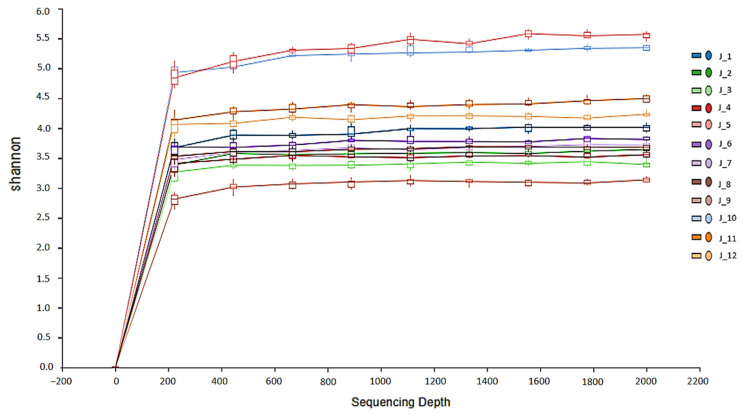
The rarefaction analysis of 12 pools of *H. verbana*. The alpha rarefaction analysis confirmed that the sequencing depth is adequate for further analysis, as shown by the parallel lines in the plot.

**Figure 3 microorganisms-13-00918-f003:**
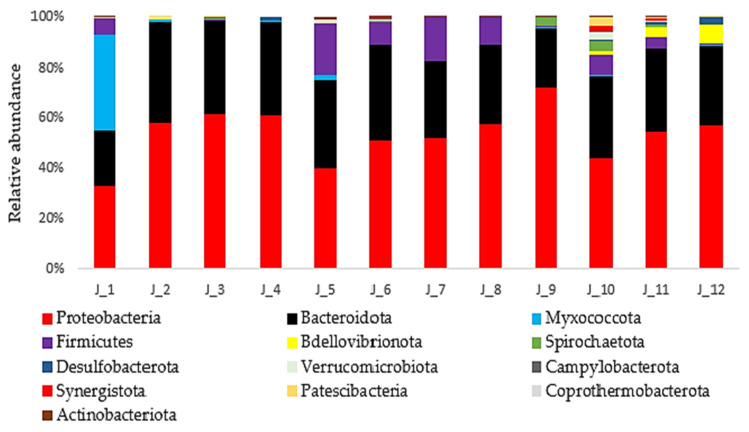
The relative abundance (%) of bacterial taxa at the phylum level from 12 pools of *H. verbana* (only 13 most abundant phyla are shown).

**Figure 4 microorganisms-13-00918-f004:**
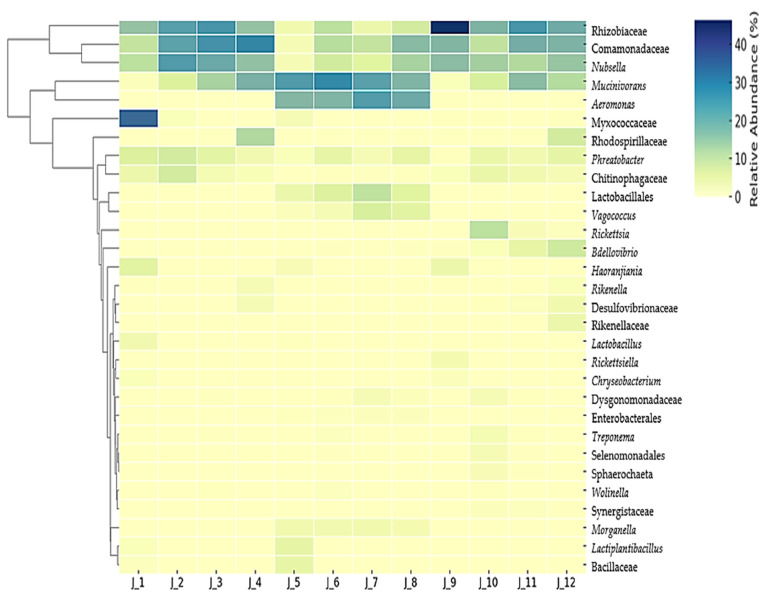
A heatmap of species abundance clustering. This heatmap illustrates the hierarchical clustering of samples based on the relative abundance of the 30 most bacterial taxa identified in region-specific segments of the 12 pools of the *H. verbana* digestive tract. The color scale represents relative abundance values, with lighter shades indicating lower abundance and darker shades indicating higher abundance.

**Figure 5 microorganisms-13-00918-f005:**
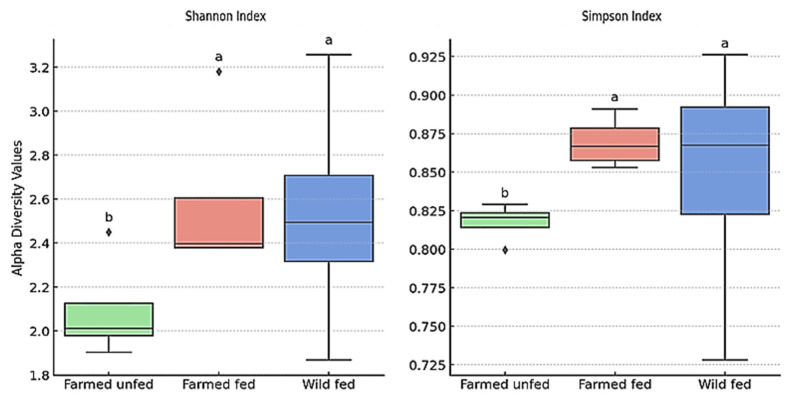
Alpha diversity based on Shannon and Simpson indexes across 4 pooled samples from each of 3 groups. A significant overall difference was found among groups (*p* = 0.004). According to Tukey’s HSD test (*p* < 0.05), the letters above the boxplots indicate statistically significant differences. Each box shows the interquartile range (Q1–Q3), the center line indicates the median, and whiskers show the minimum and maximum. For further details on the experimental design and pooling method used to generate these data, please refer to the study design section.

**Figure 6 microorganisms-13-00918-f006:**
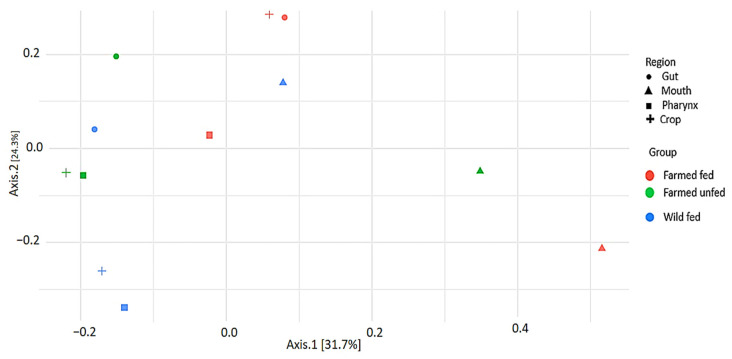
Principal coordinate analysis (PCoA) showing beta diversity of bacterial taxa among leech groups based on Bray–Curtis distance.

**Figure 7 microorganisms-13-00918-f007:**
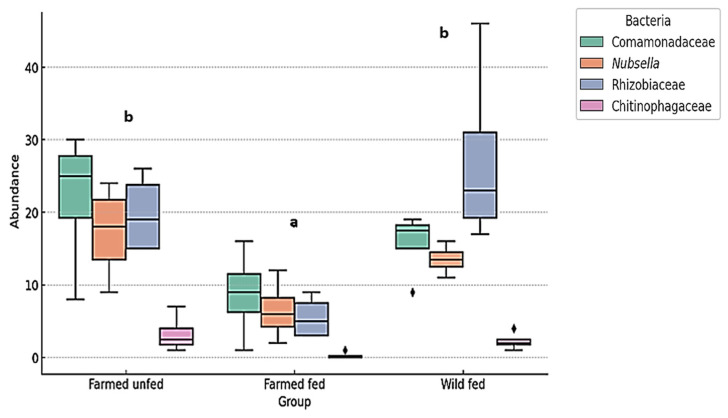
The relative abundance (%) of the most prevalent environmental bacteria in farmed fed, farmed unfed, and wild fed leeches. A boxplot representation of environmental bacterial abundances in different groups of leeches. Significant differences between groups were determined using the Kruskal–Wallis test (H = 12.18, *p* = 0.00226). Pairwise Wilcoxon rank sum tests with Bonferroni correction identified statistically significant differences between farmed unfed and farmed fed leeches (*p* = 0.00285) and between farmed fed and wild fed leeches (*p* = 0.00304), while no significant difference was found between farmed unfed and wild fed leeches (*p* = 0.5973). The letters above the boxplots indicate statistically distinct groups (*p* < 0.05). Groups sharing the same letter are not significantly different.

**Figure 8 microorganisms-13-00918-f008:**
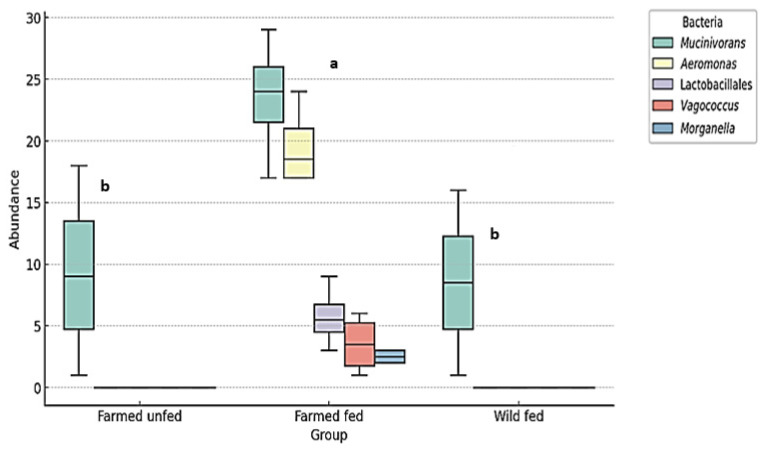
The relative abundance (%) of the most prevalent symbiotic/probiotic bacteria in farmed fed, farmed unfed, and wild fed *H. verbana*. A boxplot representation of probiotic bacterial abundances in different groups of leeches. Significant differences between groups were determined using the Kruskal–Wallis test (H = 14.74, *p* = 0.00063). Pairwise Wilcoxon rank sum tests with Bonferroni correction identified statistically significant differences between farmed unfed and farmed fed leeches (*p* = 5.81 × 10^−6^) and between farmed fed and wild fed leeches (*p* = 3.89 × 10^−6^), while no significant difference was found between farmed unfed and wild fed leeches (*p* = 0.98). The letters above the boxplots indicate statistically distinct groups (*p* < 0.05). Groups sharing the same letter are not significantly different.

**Figure 9 microorganisms-13-00918-f009:**
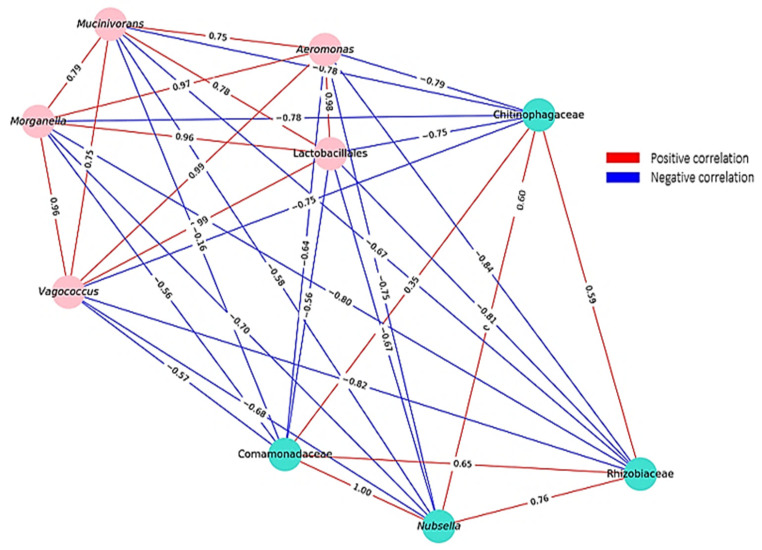
Bacterial interaction network based on Spearman correlation coefficients. Nodes represent bacterial taxa, with colors distinguishing different ecological groups: the turquoise cluster represents environmental bacteria, while the pink cluster consists of symbiotic/probiotic bacteria. Edges indicate correlations, where red represents positive correlations, and blue represents negative correlations. The edge labels display the Spearman correlation coefficients.

**Figure 10 microorganisms-13-00918-f010:**
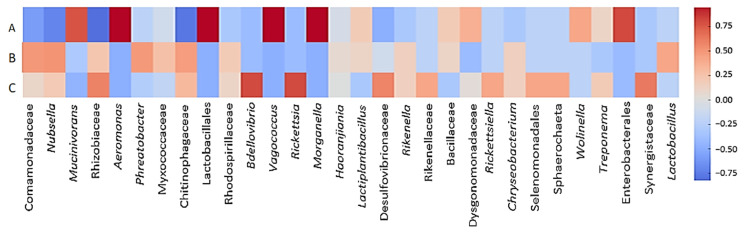
Spearman correlation between host factors (A, B, C) and bacterial abundance. (A) Farmed leeches fed on cattle blood. (B) Farmed unfed leeches. (C) Wild leeches fed on amphibian blood. The heatmap illustrates the correlation between different host-related factors and the relative abundance of bacterial genera/families in the digestive tract of *H. verbana*. Red indicates positive correlations, while blue represents negative correlations. The intensity of the color corresponds to the strength of the correlation.

**Table 1 microorganisms-13-00918-t001:** Pool names and sample types from study.

Pool No.	Main Pool	Type of Sample
J1	Farmed unfed mouth	Tissue
J2	Farmed unfed pharynx	Tissue
J3	Farmed unfed crop	Tissue
J4	Farmed unfed intestine	Tissue
J5	Farmed fed mouth	Tissue
J6	Farmed fed pharynx	ILF + Tissue
J7	Farmed fed crop	ILF
J8	Farmed fed intestine	ILF
J9	Wild fed mouth	Tissue
J10	Wild fed pharynx	ILF + Tissue
J11	Wild fed crop	ILF
J12	Wild fed intestine	ILF

**Table 2 microorganisms-13-00918-t002:** Shannon and Simpson diversity indices of microbial communities in leeches under different feeding conditions (farmed fed, farmed unfed, and wild fed) and across different anatomical regions (mouth, pharynx, crop, intestine). Different letters indicate statistically significant differences among groups based on Tukey’s test (*p* < 0.05).

Group	Region	Shannon Index (Mean ± SD)	Simpson Index (Mean ± SD)
Farmed Unfed	Mouth	2.45 ± 0.12 ^a^	0.829 ± 0.017 ^a^
	Pharynx	2.02 ± 0.10 ^ab^	0.819 ± 0.016 ^ab^
	Crop	1.90 ± 0.10 ^b^	0.800 ± 0.016 ^b^
	Intestine	2.00 ± 0.10 ^ab^	0.822 ± 0.016 ^ab^
Farmed Fed	Mouth	3.18 ± 0.16 ^c^	0.891 ± 0.018 ^c^
	Pharynx	2.41 ± 0.12 ^abc^	0.853 ± 0.017 ^abc^
	Crop	2.38 ± 0.12 ^abc^	0.859 ± 0.017 ^abc^
	Intestine	2.38 ± 0.12 ^abc^	0.874 ± 0.017 ^abc^
Wild Fed	Mouth	1.87 ± 0.09 ^d^	0.728 ± 0.015 ^d^
	Pharynx	3.26 ± 0.16 ^cd^	0.926 ± 0.019 ^cd^
	Crop	2.52 ± 0.13 ^bc^	0.854 ± 0.017 ^bc^
	Intestine	2.46 ± 0.12 ^bc^	0.881 ± 0.018 ^bc^

**Table 3 microorganisms-13-00918-t003:** Relative abundance (%) of opportunistic pathogenic bacteria in *H. verbana*.

Pathogens	Farmed Unfed	Farmed Fed	Wild Fed
*Aeromonas*	0.08 (M)	17.30 (M); 17.11 (F); 24.10 (C); 20.11 (I)	0,42 (M); 0.02 (F) 0.03 (C)
*Pseudomonas*	0.42 (M);	0.41 (M); 0.14 (F); 0.64 (C); 0.33 (I)	0.09 (M)
*Acinetobacter*	0.07 (M)	0.35 (M); 0.42 (F); 0.90 (C); 0.60 (I)	-
*Staphylococcus*	0.13 (M)	0.24 (M)	-
*Morganella*	-	3 (M); 2.11 (F); 3.02 (C); 2.14 (I)	-
*Enterococcus*	-	0.53 (M)	-
*Streptococcus*	-	0.32 (M)	-
*Rickettsia*	-	0.03 (M)	9.38 (F); 1.27 (C); 1.12 (I)
*Anaplasma*	-	0.01 (M); 0.04 (F); 0.07 (C); 0.05 (I)	0.01 (C); 0.01 (I)
*Serratia*	-	-	0.04 (F); 0.07 (C); 0.02 (I)
*Clostridia*	-	-	1.20 (F); 1.12 (C)
*Porphyromonas*	-	-	0.96 (F)
*Bacteroides*		0.75 (F)	
*Fuzobacterium*	0.05 (M)	0.53 (M)	0.02 (M)
Bacillaceae	0.57 (M)	3.85 (M)	
*Treponema*	0.12 (F); 0.33 (C);	1.10 (F)	1.95 (F); 0.53 (I)

Abbreviations: M—mouth; F—pharynx; C—crop; and I—intestine.

## Data Availability

The original contributions presented in this study are included in the article/Appendix A. Further inquiries can be directed to the corresponding author.

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
