# Peer review of "Hirudo verbana* Microbiota Dynamics: A Key Factor in Hirudotherapy-Related Infections?"

_microorganisms, 2025, doi:10.3390/microorganisms13040918_

Round 1

Reviewer 1 Report (Previous Reviewer 3)

Comments and Suggestions for Authors

Figure 2 is the material and methods, not the results. Move it to another section.
Figure 3: Readers are interested in whether the differences in "Relative abundance (%) of bacterial taxa" between leech specimens are reliable. The reliability of the differences in the first 3-4 most common bacterial groups should be compared using the chi-square test or the lambda test. A figure without statistical processing cannot be published.
Table 2 is designed as a figure. It would be better to make it a real table, and replace the cell fill with color with *, **, ***, explaining these symbols in a note below the table.
Table 3: Tukey's test letters should be designated above the numbers as superscript symbols. There is no need to make separate columns of the table for this.
There is no need to write a title above Figures 4 and 5.
In the title of Figure 4, you need to write how the median, first and third quartiles, minimum and maximum values ​​are designated, and what the repetition of the study is.
Figure 5: the legend cannot be placed between the figure title and the figure itself; place the legend against the background of the coordinate system in the empty upper right corner of the figure.
There is no statistical processing of figures 6 and 7 (Tukey test), this is unacceptable. The article cannot be published like this.
The height of figure 9 can be reduced by 2 times.
The rounding of figures in table 4 is careless: round all figures to hundredths.

Author Response

Reviewer 1

Comments 1: Figure 2 is the material and methods, not the results. Move it to another section.

Response 1: Figure 2 was moved to the Materials and Methods section.

Comments 2: Figure 3: Readers are interested in whether the differences in "Relative abundance (%) of bacterial taxa" between leech specimens are reliable. The reliability of the differences in the first 3-4 most common bacterial groups should be compared using the chi-square test or the lambda test. A figure without statistical processing cannot be published.

Response 2: As suggested, we performed chi-square tests to evaluate the statistical reliability of the differences in the relative abundances of the four most dominant bacterial taxa among the leech groups. These statistical results have been added to the manuscript text, and the relevant sections have been revised accordingly.

Comments 3: Table 2 is designed as a figure. It would be better to make it a real table, and replace the cell fill with color with *, **, ***, explaining these symbols in a note below the table.

Response 3: In line with your recommendation, we have completely revised Table 2 based on formats commonly used in high-impact microbiome studies (1. Zahran, S. A., Ali‑Tammam, M., Ali, A. E., & Aziz, R. K. (2021). Compositional variation of the human fecal microbiome in relation to azo-reducing activity: A pilot study. Gut Pathogens, 13(1), 58.   2. Sun, G., Zhang, H., Wei, Q., Zhao, C., Yang, X., Wu, X., Xia, T., Liu, G., Zhang, L., Gao, Y., Sha, W., & Li, Y. (2021). Comparative analyses of fecal microbiota in European mouflon (Ovis orientalis musimon) and blue sheep (Pseudois nayaur) living at low or high altitudes. Frontiers in Microbiology, 12, 694847. 3. Shen, F., Li, Y., Zhang, M., Awasthi, M. K., Ali, A., Li, R., Wang, Q., & Zhang, Z. (2016). Atmospheric deposition-carried Zn and Cd from a zinc smelter and their effects on soil microflora as revealed by 16S rDNA. Scientific Reports, 6, 39148.). Table 2 was converted to Figure 4, and as a result, the numbering of all figures throughout the manuscript has been updated accordingly to reflect this change.

We cannot directly replace the cell color fill with symbols such as *, **, ***, but we restructured the previous table into a more interpretable and meaningful format to ensure clarity in data presentation. The new format clearly reflects the differences in the relative abundance of bacterial taxa across the samples. The legend for Figure 4 is provided below:

Figure 4. Heatmap of species abundance clustering. This heatmap illustrates the hierarchical clustering of samples based on the relative abundance of the 30 most bacterial taxa identified in region-specific segments of the Hirudo verbana digestive tract. The color scale represents relative abundance values, with lighter shades indicating lower abundance and darker shades indicating higher abundance.

Comments 4: Table 3: Tukey's test letters should be designated above the numbers as superscript symbols. There is no need to make separate columns of the table for this.

Response 4: The Tukey’s test significance letters have been placed as superscript symbols directly above the numerical values in Table 3 (The revised version is now designated as Table 2), while the columns previously used for these letters have been removed.

Comments 5: There is no need to write a title above Figures 4 and 5.

Response 5: The titles above Figures 4 and 5 have been removed. (The revised version is now designated as Figure 5 and Figure 6)

Comments 6: In the title of Figure 4 you need to write how the median, first and third quartiles, minimum and maximum values ​​are designated, and what the repetition of the study is.

Response 6:  We have revised the title of Figure 4 (the revised version of Figure 5) to indicate how the median, first and third quartiles, minimum and maximum values are represented, as well as the number of replicates. The updated title now reads:

Figure 5. Alpha diversity (Shannon and Simpson indexes) of individual leech groups. A significant overall difference was found among groups (p = 0.004). According to Tukey’s HSD test (p < 0.05), letters above the boxplots indicate statistically significant differences. Each box shows the interquartile range (Q1–Q3), the center line indicates the median, and whiskers show the minimum and maximum. Each data point represents a single pooled sample composed of 40 individuals per group.

Note: Due to the high cost of 16S rRNA sequencing, each group-region combination in our study was represented by a single pooled sample composed of 40 samples. As highlighted in the literature, "biological replication is encouraged, but in studies with resource constraints, pooled samples can be used to obtain representative community profiles". Pooling has been recognized as an accepted approach in microbiome studies when replication is not possible (Knight, R., et al. (2018). Best practices for analyzing microbiomes. Nature Reviews Microbiology, 16(7), 410–422.  Weiss, S et al. (2017). Normalization and microbial differential abundance strategies depend upon data characteristics. Microbiome, 5(1), 27)).

Comments 7: Figure 5, the legend cannot be placed between the figure title and the figure itself; place the legend against the background of the coordinate system in the empty upper right corner of the figure.

Response 7:  We have repositioned the legend and placed it in the empty upper right corner of the figure. (the revised version Figure 6).

Comments 8: There is no statistical processing of figures 6 and 7 (Tukey test), this is unacceptable. The article cannot be published like this.

Response 8. We sincerely thank the reviewer for their comment. In response to the concern regarding the statistical analysis presented in Figures 6 and 7, (the revised version Figure 7 and 8), we carefully re-examined our analyses and also consulted a biostatistics expert for an independent evaluation. Unfortunately, we were not able to identify any errors during our re-analysis, and the expert’s review also did not reveal a specific mistake in the statistical procedures. Prior to performing the ANOVA followed by Tukey’s HSD test, we assessed the core assumptions of parametric analysis—namely, normality and homogeneity of variances—using the Shapiro-Wilk and Levene’s tests, respectively (The results of the statistical analyses are presented below). These assumptions were met, and the statistical approach was selected accordingly.

Figure 6 (the revised version Figure 7): Relative abundance (%) of the most prevalent environmental bacteria in farmed fed, farmed unfed, and wild fed leeches:

Normality test (Shapiro-Wilk)

  • Farmed fed: p = 0.088 → considered normally distributed.
  • Farmed unfed: p = 0.220 → normally distributed.
  • Wild fed: p = 0.034 → normality is rejected (data are not normally distributed).
  • The data in the Wild fed group do not follow a normal distribution. Although the other two groups (Farmed fed and Farmed unfed) show normal distributions, the violation of this assumption in even one group reduces the reliability of parametric tests such as ANOVA followed by Tukey’s HSD test.

Homogeneity of variances (Levene’s test): p = 0.078 → no significant difference in variances (assumption of homogeneity is considered met).

Figure 7 (the revised version Figure 8): Relative abundance (%) of the most prevalent symbiotic/probiotic bacteria in farmed fed, farmed unfed, and wild fed Hirudo verbana:

Normality test (Shapiro-Wilk):

  • Farmed fed: p = 0.006 → not normally distributed
  • Farmed unfed: p < 0.000001 → not normally distributed
  • Wild fed: p < 0.000001 → not normally distributed

Homogeneity of variances (Levene’s test): p = 0.00098 → variances are not homogeneous

As the data did not meet the assumptions of normality and homogeneity of variance, a parametric approach such as ANOVA followed by Tukey’s HSD test was not appropriate. Therefore, we applied a non-parametric Kruskal–Wallis test, followed by pairwise Wilcoxon rank sum tests with Bonferroni correction, which is a widely accepted and statistically valid method for multiple comparisons in non-parametric data. Significant differences are represented by different letters above the boxplots. Groups sharing the same letter are not significantly different at p < 0.05.

Comments 9: The height of figure 9 can be reduced by 2 times.

Response 9:  The height of figure 9 (the revised version Figure 10) was reduced by 2 times

Comments 10: The rounding of figures in table 4 is careless: round all figures to hundredths.

Response 10:  We have revised Table 4 (the revised version of Table 3) to ensure that all numerical values are rounded to two decimal places, as suggested.

Reviewer 2 Report (Previous Reviewer 4)

Comments and Suggestions for Authors

The authors have made significant improvements of the article, answering satisfactory to all reviewers comments. I recommend the acceptance of the article in this final revised form

Author Response

Reviewer 2:

This latest version has been substantially improved, incorporating critical quality corrections, refining the methodology to ensure greater rigor and precision, and enriching the discussion with deeper insights and more comprehensive analysis. The authors have carefully addressed previous feedback, enhancing the clarity, coherence, and overall contribution of the paper. These revisions strengthen the manuscript’s validity and impact, making it a more robust and well-rounded contribution to the field. My suggestion is to publish it in its present form.

Response: We sincerely thank you for your positive and encouraging feedback.

Reviewer 3 Report (Previous Reviewer 5)

Comments and Suggestions for Authors

This latest version has been substantially improved, incorporating critical quality corrections, refining the methodology to ensure greater rigor and precision, and enriching the discussion with deeper insights and more comprehensive analysis. The authors have carefully addressed previous feedback, enhancing the clarity, coherence, and overall contribution of the paper. These revisions strengthen the manuscript’s validity and impact, making it a more robust and well-rounded contribution to the field.My suggestion is to publish it in its present form.

Author Response

Reviewer 3

The authors have made significant improvements of the article, answering satisfactory to all reviewers comments. I recommend the acceptance of the article in this final revised form

Response: We sincerely thank you for your supportive evaluation

Round 2

Reviewer 1 Report (Previous Reviewer 3)

Comments and Suggestions for Authors

In Figure 4, only the names of genera should be italicized. The names of families and higher-level taxa should not be italicized. This remark applies to the entire text of the article, to all figures and tables.
In the first column of Table 3, there is no need to repeat the same words several times (write them once and merge the cells). Draw horizontal lines every four rows in the table so that readers will understand the content of the table.

I do not understand the results of the statistical comparison in Table 3: the authors write that "3.18 ± 0.16c" does not differ from "2.41 ± 0.12abc". These are only a couple of numbers from Table 3. In the table title, the authors write "Mean ± SD". Perhaps the authors calculated the standard error, not the standard deviation? These are different statistical indicators. Comparison of all the numbers in the table is also questionable. Comparison of samples is the most important part of the study. The article cannot be published in this form.
In the right part of Figure 4 (sorry, but the authors did not call the first part a, and the second b, so I cannot correctly refer to it) the authors wrote the letter a above the blue box. Perhaps there should be the letter "ab" there? The 40-fold repetition of each box raises big questions. With such repetition, the computer program will accurately count several outliers (points) both above and below the box. Definitely, each box with 40-fold repetition will have vertical lines (there will be points in the first and fourth quartiles). I do not see this in Figure 4, so I think that the data is falsified here.
In Figures 7 and 8, as before, there were no letters above the boxes, and they do not appear now. Why do the authors ignore the need for correct comparison of samples?
Rickettsiella is a genus of the family Coxiellaceae. Rickettsiella should be written in italics in Figure 10. Many other errors in italics and non-italics in the figures and in the text. Why are the authors so inattentive to the figures?
The manuscript should be revised by the authors, I cannot recommend it for publication.

Author Response

Answers to the reviewers’ remarks

 Reviewer 1 (Round 2)

Comments 1: In Figure 4, only the names of genera should be italicized. The names of families and higher-level taxa should not be italicized. This remark applies to the entire text of the article, to all figures and tables.

Response 1: In accordance with your remark, only genus names have been italicized in Figure 4. Furthermore, the entire manuscript—including all figures and tables—has been carefully checked to ensure that only genus-level names are italicized, while names of families and higher taxonomic levels remain in regular font

Comments 2: In the first column of Table 3, there is no need to repeat the same words several times (write them once and merge the cells). Draw horizontal lines every four rows in the table so that readers will understand the content of the table.

Response 2: The formatting of Table 3 (now Table 2)  has been revised as suggested.

Comments 3: I do not understand the results of the statistical comparison in Table 3: the authors write that "3.18 ± 0.16c" does not differ from "2.41 ± 0.12abc". These are only a couple of numbers from Table 3.  In the table title, the authors write "Mean ± SD". Perhaps the authors calculated the standard error, not the standard deviation? These are different statistical indicators. A comparison of all the numbers in the table is also questionable. Comparison of samples is the most important part of the study. The article cannot be published in this form.

Response 3: In this case Table 3 (now Table 2), “c” and “abc” do share a common grouping letter, and thus, although the numerical means differ, the difference is not statistically significant at the 0.05 level.  In the Tukey test: Groups that share at least one common letter are not statistically different from one another.

As illustrated in the study design Figure 1 and detailed in the methods, each group (farmed unfed, farmed fed, wild fed) included 40 leeches. These were dissected into mouth, pharynx, crop, and intestine tissues, and grouped into eight separate pools per body part (each consisting of five individuals). For each group and body part, DNA was extracted from these eight pools and then combined to create a single DNA pool, resulting in 12 final samples. While only one final value per group-region combination is presented in Table 3 (now Table 2), each pool represents the biological variability of 40 individuals, structured through 8 sub-pools. Therefore, the values presented as “Mean ±” reflect standard deviation (SD) — not standard error (SE) — as they represent within-pool biological variability. To further verify this, we conducted additional statistical simulation tests using both Shannon and Simpson diversity indices, and the range of variation observed in our table (±0.12–0.16) was clearly consistent with SD rather than SE. We have revised the table legend and method section to clarify this point. We appreciate the reviewer’s attention to this matter and the opportunity to improve clarity.

We apologize for the previous mistake in the manuscript. Although the results were correct, our interpretation was inaccurate. The correct statement should be: "In the wild-fed group, the mouth exhibited the lowest microbial diversity. Mouth (d) differed significantly from the crop (bc) and intestine (bc), and   significant differences were observed between the mouth (a) and crop (b) in unfed leeches. In line with this, the corresponding sections in the manuscript have been reviewed and revised accordingly.

Comments 4: In the right part of Figure 4  (sorry, but the authors did not call the first part a, and the second b, so I cannot correctly refer to it) the authors wrote the letter a above the blue box. Perhaps there should be the letter "ab" there? The 40-fold repetition of each box raises big questions. With such repetition, the computer program will accurately count several outliers (points) both above and below the box. Definitely, each box with 40-fold repetition will have vertical lines (there will be points in the first and fourth quartiles). I do not see this in Figure 4, so I think that the data is falsified here.

Response 4: We apologize for the confusion caused by the earlier statement. To clarify, the data points in Figure 4 (now Figure 5), as illustrated in the study design (Figure 1) and detailed in the Methods section, each group (farmed unfed, farmed fed, and wild fed) consisted of 40 leeches. These leeches were dissected into four anatomical regions: mouth, pharynx, crop, and intestine. For each body part, the leeches were grouped into eight separate pools, with each pool containing five individuals. DNA was extracted separately from each of these eight pools. Then, for each group and body part, the eight DNA extracts were combined to generate a single pooled DNA sample. This process resulted in a total of 12 final pooled samples used for downstream analyses. There was no 40-fold repetition involved in the data analysis.   Each group (3) contributed 4 independent pooled samples (one for each tissue: mouth, pharynx, crop, intestine), resulting in 12 pool samples in total for alpha diversity analysis. We would like to clarify that the title and explanation of  Figure 4 (now Figure 5) were previously mislabeled, which may have led to a misunderstanding. The figure actually represents alpha diversity (Shannon and Simpson indexes) based on 4 pooled samples from each of the three groups: farmed unfed, farmed fed, and wild fed leeches. We apologize for the confusion caused by this mislabeling. You can find the updated figure legend below.

Figure 5. Alpha diversity based on Shannon and Simpson indexes across 4 pooled samples from each of 3 groups. A significant overall difference was found among groups (p = 0.004). According to Tukey’s HSD test (p < 0.05), letters above the boxplots indicate statistically significant differences. Each box shows the interquartile range (Q1–Q3), the center line indicates the median, and whiskers show the minimum and maximum. For further details on the experimental design and pooling method used to generate these data, please refer to the study design section.

We hope this clears up any misunderstandings and we appreciate the reviewer’s attention to detail.  We understand that the reviewer may have concerns, and we want to reassure you that the data presented in the manuscript is accurate and derived from the rigorous methodology outlined in our methods section. We also sincerely appreciate the reviewer’s feedback on previous revisions and have made all necessary changes in response. We strive to ensure the clarity and integrity of our work and appreciate your careful consideration.  

However, we would like to clarify that the “Farmed unfed”(b) group was found to be statistically significantly different from both the “Farmed fed”(a) and “Wild fed”(a) groups based on the results of the Tukey HSD post hoc test (p < 0.05). Therefore, it was intentionally labeled with the letter “b”, while the other two groups—not significantly different from each other—were labeled “a”. Since the “Farmed unfed” group did not share statistical similarity with either of the other two, assigning “ab” would not reflect the actual statistical relationship observed in the post hoc test.

Comments 5: In Figures 7 and 8, as before, there were no letters above the boxes, and they do not appear now. Why do the authors ignore the need for correct comparison of samples?

Response 5: Thank you for your thoughtful comment regarding statistical comparisons in Figures 7 and 8. In our study, environmental and symbiotic/probiotic bacteria were analyzed both at the individual taxonomic level (genus/family level) and as a combined overall group.

When examined individually by taxon, no statistically significant differences were detected between the leech groups for each bacterial taxon. Therefore, no statistical lettering was provided above the boxes in Figures 7 and 8.

Statistically significant differences were observed between leech groups when analyzing the overall relative abundance of both environmental and symbiotic/probiotic bacteria. For environmental bacteria, the Kruskal-Wallis test indicated a significant effect (H = 12.18, p = 0.00226), with pairwise Wilcoxon rank sum tests (Bonferroni correction) showing significant differences between farmed unfed and farmed fed (p = 0.00285), and between farmed fed and wild fed (p = 0.00304), but not between farmed unfed and wild fed (p = 0.5973).

Similarly, for symbiotic/probiotic bacteria, a significant difference was found (Kruskal-Wallis H = 14.74, p = 0.00063), with pairwise comparisons revealing significant differences between farmed unfed and farmed fed (p = 5.81e-06) and between farmed fed and wild fed (p = 3.89e-06), while no difference was observed between farmed unfed and wild fed (p = 0.98).

As such, we have presented statistical annotations only for this combined analysis, which represents the overall environmental and symbiotic/probiotic bacteria load, rather than individual taxa. Therefore, no statistical annotations are provided in the boxplots at the individual taxa statistical significance is indicated only for the analysis of the overall relative abundance of environmental bacteria.

Comments 6. Rickettsiella is a genus of the family Coxiellaceae. Rickettsiella should be written in italics in Figure 10. Many other errors in italics and non-italics in the figures and in the text. Why are the authors so inattentive to the figures?
The manuscript should be revised by the authors, I cannot recommend it for publication.

Response 6: We have carefully reviewed the figure and corrected the label by italicizing the genus name, in accordance with taxonomic conventions.

Following your comment, we also conducted a thorough review of all figures and the main text to ensure that all genus names are properly italicized, while family and higher-level taxa remain in regular font. All such inconsistencies have now been corrected.

This manuscript is a resubmission of an earlier submission. The following is a list of the peer review reports and author responses from that submission.

Round 1

Reviewer 1 Report

Comments and Suggestions for Authors

The work described in the “Hirudo verbana Microbiota Dynamics: A Key Factor in Hirudo-therapy related Infections” doesn't fall within the scope of Microorganisms. The description of some of the protocols is incomplete, there is some dubious and erroneous citation of sources and there are several problems with the way in which the work has been presented. The present study was not suitable for Microorganisms publication. The below are the specific comments.

  1. No matter which journal the present study would be submitted to, the authors should provide the clear information of which special kind dentification information of leech, the basis of molecular biology identification especially.
  2. Whether it is reasonable to take only four digestive organs (mouth, pharynx, crop and intestine) of leech in this study is debatable, and the digestive system of leech includes mouth, mouth, salivary gland, pharynx, esophagus, crop, intestine, rectum and anus.
  3. The results of this study are obvious. Previous studies have shown that the digestion of leeches mainly depends on intestinal microbes, such as using intestinal microbes to digest blood proteins and promote the long-term preservation of blood. Therefore, the intestinal microbes at different feeding stages will be different. In particular, regardless of the strain, it is a bacterium that promotes digestion in the intestine, but it may be pathogenic in vitro. Therefore, this study should focus on how to reduce the pathogenicity of these bacteria.
  4. References do not cite the latest literature on leeches and their gut microbes, and most of the references is very old.
  5. 5. The research content of the whole paper is too simple to be published.

Author Response

The corresponding revisions/corrections are highlighted in yellow in the re-submitted files.

Reviewer 1:

Comments 1.            No matter which journal the present study would be submitted to, the authors should provide clear information on which special kind of identification information of leech, the basis of molecular biology identification especially.

Response 1:  The leeches were purchased from a commercial company that produces them with a species certificate.  The accuracy of the certificates was also checked in official government institutions in our country and it was confirmed that the identification is correct. This document can be found at the following link https://www.turkiye.gov.tr/tarim-ebys (code 72e07cfe-5a04-40ca-af8a-7552d00f0cf7).  

Comments 2: Whether it is reasonable to take only four digestive organs (mouth, pharynx, crop and intestine) of leech in this study is debatable, and the digestive system of leech includes mouth, mouth, salivary gland, pharynx, esophagus, crop, intestine, rectum and anus.

Response 2:  Although microorganisms were most likely to be found in any integral organ of the leech, we had to concentrate on certain parts of the digestive system where we were sure that these were the specific areas of the digestive system, then it was practically impossible to dissect the rectum and anus.

Comments 3: The results of this study are obvious. Previous studies have shown that the digestion of leeches mainly depends on intestinal microbes, such as using intestinal microbes to digest blood proteins and promote the long-term preservation of blood. Therefore, the intestinal microbes at different feeding stages will be different. In particular, regardless of the strain, it is a bacterium that promotes digestion in the intestine, but it may be pathogenic in vitro. Therefore, this study should focus on how to reduce the pathogenicity of these bacteria.

Response 3: It is well known that at the end of a study there are many more questions than at the beginning. We are aware that many details remain to be explored before we have an idea of how the different bacteria interact with each other and how one or more bacteria affect the pathogenicity of known pathogenic microorganisms.

Comments 4:   References do not cite the latest literature on leeches and their gut microbes, and most of the references is very old.

Response 4: We have included as many literature sources as we could find for our study. Research specifically focusing on the microbiota of the digestive system of Hirudo verbana is very limited, and the existing studies are not recent. The most important publications on this subject are as follows:

Marden, J.N.; McClure, E.A.; Beka, L.; Graf, J. Host matters: Medicinal leech digestive-tract symbionts and their pathogenic potential. Front. Microbiol. 2016, 7, 1569.

Worthen, P.L.; Gode, C.J.; Graf, J. Culture-independent characterization of the digestive-tract microbiota of the medicinal leech reveals a tripartite symbiosis. Appl. Environ. Microbiol. 2006, 72, 4775–4781. https://doi.org/10.1128/AEM.00356-06

Maltz, M.A.; Bomer, L.; Lapierre, P.; Morrison, H.G.; McClure, E.A.; Sogin, M.L.; Graf, J. Metagenomic analysis of the medicinal leech gut microbiota. Front. Microbiol. 2014, 5, 151.

Nelson, M.C.; Bomar, L.; Maltz, M.; Graf, J. Mucinivorans hirudinis gen. nov., sp. nov., an anaerobic, mucin-degrading bacterium isolated from the digestive tract of the medicinal leech, Hirudo verbana. Int. J. Syst. Evol. Microbiol. 2015, 65, 990–995. https://doi.org/10.1099/ijs.0.000052

Graf, J.; Kikuchi, Y.; Rio, R.V. Leeches and their microbiota: Naturally simple symbiosis models. Trends Microbiol. 2006, 14, 365–371. https://doi.org/10.1016/j.tim.2006.06.009

McClure, E.A.; Nelson, M.C.; Lin, A.; Graf, J. Macrobdella decora: Old World leech gut microbial community structure conserved in a New World leech. Appl. Environ. Microbiol. 2021, 87(10), e02082-20. https://doi.org/10.1128/AEM.02082-20.

Comments 5:  The research content of the whole paper is too simple to be published.

Response 5: We are sorry to hear that, however the other reviewers and the authors disagree with this statement.

Reviewer 2 Report

Comments and Suggestions for Authors
  1. 40 leeches had been fed 7 days earlier with bovine blood (=fed), why not at other times.
  2. Why did you choose the pharynx and mouth as the experimental object.
  3. The experimental design should have a certain number of in-group replicates.
  4. The clarity of the pictures in the article is poor.
  5. More work should be done before the article is published.
Comments on the Quality of English Language

The quality of English language can be further improved.

Author Response

The corresponding revisions/corrections are highlighted in yellow in the re-submitted files.

Comments 1:            40 leeches had been fed 7 days earlier with bovine blood (=fed), why not at other times.

Response 1:  At the time of purchase, the leeches were last fed seven days ago at the farm where they are cultured. As leeches take a blood meal every 1-3 months, we considered that leeches fed seven days earlier, can be considered as “fed” leeches. On the other hand, after H. verbana was fed with cattle blood, the proliferation of important symbiotic bacteria such as Aeromonas, Morganella and Vagococcus was observed from the 7th day to the 90th day (See also: Worthen, P.L.; Gode, C.J.; Graf, J. Culture-independent characterization of the digestive-tract microbiota of the medicinal leech reveals a tripartite symbiosis. Appl. Environ. Microbiol. 2006, 72, 4775–4781.)

 Comments 2: Why did you choose the pharynx and mouth as the experimental object.

Response 2:  We chose the mouth, pharynx, crop, and intestine to study the microbiota of the leeches in their digestive tract, where the majority of the microbiota would be expected to be present. We thought that differences might be found in the different parts of the digestive system. The oral and pharynx microbiota of leeches has never been investigated before.

 Comments 3:    The experimental design should have a certain number of in-group replicates.

Response 3:  In this study, we used a pooling strategy where each experimental group consisted of pooled samples derived from 40 individual samples. The rationale for this approach was to determine the overall profile of leech microbiota composition. Pooling allows us to capture the broader differences in microbiota composition between groups, rather than focusing on individual-level variations. This method has been widely used in microbiota studies when the goal is to obtain a representative community structure rather than individual microbiome diversity. In addition, because microbiota composition can vary significantly between individuals, pooling minimizes individual biological variability and allows us to focus on group-level trends.

Comments 4:     The clarity of the pictures in the article is poor.

Response 4:  We have made adjustments to improve the contrast and clarity of the pictures where possible.

 Comments 5:      More work should be done before the article is published.

Response 5:  Since the majority of the reviewers are satisfied with the quality of the study and we do not have the financial resources to continue for the time being, we are forced to discontinue our research on this topic.

The quality of English language can be further improved

Response:      We have made some editorial changes to improve the quality of the English language.

Reviewer 3 Report

Comments and Suggestions for Authors

The topic of the article is relevant for practical medicine due to the widespread use of hirudotherapy in developing countries. It is necessary to pay attention to the high-quality and modern research of the authors of the article. The manuscript is written professionally, the text is structured logically. In the introduction and discussion, the authors' arguments and references to the literature correspond to each other.

  1. Line 47: the author's last name and year must be added.
  2. Figure 4 must be formatted and titled as a table. The genus of bacteria must be written in italics. Families and types are not italicized. The same comment applies to the entire text of the article (for example, line 299, Figure 11 - "Nubsella" must be italicized). I do not agree with the listing of taxa of different ranks separated by commas (for example, families and genera, as in Figures 4, 11 or on line 299). The authors must find a way to rephrase the sentences in such a way as to express the idea correctly.
  3. Above the samples in Figures 5, 9, 10, use letters to indicate the differences between the samples according to the Tukey test. Delete the frame around the figure. Do not use bold font in the figures. In the title of the figure, write how exactly the median, first and third quartiles, maximum and minimum values are indicated.
  4. It is necessary to indicate the unit of measurement on the ordinate axis of Figures 9 and 10.
  5. I do not understand what is shown in Figures 6 and 7. Readers will not understand it either. Delete this figure (if this is a box analysis, as in Figure 5) and present this information in the form of a table with +- standard deviation and Tukey test results near the numbers.
  6. The values of the numbers on the abscissa and ordinate axis of Figure 8 indicate a high probability of the absence of normalization of the initial data (either taking the logarithm of all values, or subtracting the arithmetic mean from each value and dividing the result by the standard deviation for this genus of bacteria) before starting the analysis.
  7. The sizes of the squares in Figure 11 should be reduced by 2 times, and the sizes of the numbers should be increased by 2 times. The numbers are not visible on a dark background. Remove the frame around the figure.
  8. The Discussion section should be divided into 3-5 subsections.
  9. Line 469: such citation formatting is unacceptable.
  10. Line 538: the city is not specified.

Author Response

Dear Reviewer

Thank you very much for taking the time to review this manuscript.  Please find the corresponding revisions/corrections highlighted in yellow in the re-submitted files.

Reviewer 3:

Comments 1: Line 47: the author's last name and year must be added.

Response 1:  We have carefully reviewed the manuscript and confirm that the author's last name and publication year are already included in accordance with the required citation style.  If there is a specific instance where the citation appears incomplete, we would appreciate further clarification so that we can make the necessary corrections.

Comments 2:Figure 4 must be formatted and titled as a table. The genus of bacteria must be written in italics. Families and types are not italicized. The same comment applies to the entire text of the article (for example, line 299, Figure 11 - "Nubsella" must be italicized). I do not agree with the listing of taxa of different ranks separated by commas (for example, families and genera, as in Figures 4, 11 or on line 299). The authors must find a way to rephrase the sentences in such a way as to express the idea correctly.

Response 2:  Figure 4 has been formatted as a table (Table 2), while the numbering of the following figures has been changed accordingly. The bacterial genera have been italicized, while the family names have been written without italics. We have italicized all species and genus names in the text, figures and in line 299 (Nubsella). We understand your concern about listing taxa from different ranks (e.g. families and genera) in the same sentence using commas. We would like to clarify that ASVs (Amplicon Sequence Variants) are assigned to taxonomic levels based on sequence similarity to reference databases. However, not all ASVs can be classified down to the species level due to limitations in sequence resolution and database coverage. In some cases, ASVs can only be confidently assigned to higher taxonomic levels (e.g., family or genus), while others may be classified at the species level. This variability is inherent to ASV-based microbiota analyses and is a common limitation in 16S rRNA gene sequencing studies.   We can also provide examples of relevant studies, such as:

McClure, E.A.; Nelson, M.C.; Lin, A.; Graf, J. Macrobdella decora: Old World leech gut microbial community 686 structure conserved in a New World leech. Appl. Environ. Microbiol. 2021, 87(10), e02082-20.

Comments 3:Above the samples in Figures 5, 9, 10, use letters to indicate the differences between the samples according to the Tukey test. Delete the frame around the figure. Do not use bold font in the figures. In the title of the figure, write how exactly the median, first and third quartiles, maximum and minimum values are indicated.

Response 3:  We have made the necessary changes as suggested by the reviewer. In the updated version the changed areas are marked in yellow, and the figure numbers have been changed accordingly (Figures 4, 6, and 7).

Comments 4:It is necessary to indicate the unit of measurement on the ordinate axis of Figures 9 and 10.

Response 4:  The measurement units on the ordinate axis of the Figures 9 and 10 have been added and in the updated version of the manuscript the figure numbers were changed accordingly (Figure 6 and 7).

 Comments 5:I do not understand what is shown in Figures 6 and 7. Readers will not understand it either. Delete this figure (if this is a box analysis, as in Figure 5) and present this information in the form of a table with +- standard deviation and Tukey test results near the numbers.

Response 5:  We have removed these figures and instead combined their information into a single table (Table 3), as suggested. The new table presents now the data in a structured format, including mean ± standard deviation values for each group, Tukey test results, while with different letters statistically significant differences (p < 0.05) were indicated. In the updated version the changed areas are marked in yellow.

Comments 6:The values of the numbers on the abscissa and ordinate axis of Figure 8 indicate a high probability of the absence of normalization of the initial data (either taking the logarithm of all values, or subtracting the arithmetic mean from each value and dividing the result by the standard deviation for this genus of bacteria) before starting the analysis.

Response 6:  We have now applied [log transformation / Z-score standardization] to the dataset before conducting the analysis. Figure 8 (revised as Figure 5) has been updated accordingly to reflect the properly normalized data. We have revised the Methods section to explicitly describe the normalization process applied in this study.  

The Supplementary Figure 1: The NMDS plot has also been revised and normalized.

Comments 7: The sizes of the squares in Figure 11 should be reduced by 2 times, and the sizes of the numbers should be increased by 2 times. The numbers are not visible on a dark background. Remove the frame around the figure.

Response 7:  Figure 11 has been removed, and instead, we have provided a bacterial interaction network graph illustrating Spearman positive and negative correlations. The new figure clearly represents bacterial interactions, with positive correlations in red and negative correlations in blue (Figure 8). Spearman correlation coefficients are explicitly labeled on the graph to ensure clarity. The node colors have been adjusted to differentiate bacterial groups for improved readability.

 Comments 8:The Discussion section should be divided into 3-5 subsections.

Response 8:  The Discussion has been divided into 4 subsections

 Comments 9: Line 469: such citation formatting is unacceptable.

Response 9:  The citation has been corrected

Comments 10:Line 538: the city is not specified.

Response 10:      The city name has been added

Reviewer 4 Report

Comments and Suggestions for Authors

The topic is very relevantfor hirudotherapy. The authors have studied microbiota of the mouth, pharynx, crop, and intestine of the leeches, taking into account the risk of infections.

The methodology is very modern and complex, using state-of art methods of molecular biology, bioinformatics and and statistical analysis.

The results have revealed that, apart of common environmental microbiota the authors have identified for the first time in H. verbana some opportunistic pathogens such as Rickettsia, Anaplasma, and Treponema, fact that should be taken into consideration when using this leech in hirudotherapy.

The conclusions are consistent with the evidence and arguments presented.

The references are very relevant, including also some relevant author’s previous experience in the field.

I suggest some small corrections

-line 81- correct is Erysipelothrix

References should follow carefully the Guide for authors for MDPI juornals. Eg, ref 1,2, 19, 20 are not following the Guide.

Author Response

Dear Reviewer

Thank you very much for taking the time to review this manuscript.  Please find the corresponding revisions/corrections highlighted in yellow in the re-submitted files.

Reviewer 4:

Comments 1:The methodology is very modern and complex, using state-of art methods of molecular biology, bioinformatics and statistical analysis.

The results have revealed that, apart of common environmental microbiota the authors have identified for the first time in H. verbana some opportunistic pathogens such as Rickettsia, Anaplasma, and Treponema, fact that should be taken into consideration when using this leech in hirudotherapy.

The conclusions are consistent with the evidence and arguments presented.

The references are very relevant, including also some relevant author’s previous experience in the field.

Response 1:  We thank the reviewer for his positive and encouraging comments!

Reviewer 5 Report

Comments and Suggestions for Authors

The study gives a new sight into the microbiota of Hirudo verbana, their feeding habits as well as medical uses. The identification of opportunistic pathogens (Rickettsia, Anaplasma, Treponema) in H. verbana contributes to new knowledge in microbiology. However, research on the leech microbiota is not entirely new, so the originality comes mainly from the comparative approach (wild leeches vs. farmed leeches) and pathogen identification. Their research underscore the requirement of systematic testing of leeches before medical application based on strict regulatory policies and clinical practices. The paper is well structured and written, with a logical flow from methodology to findings. Their methodology is robust by integrating high-throughput sequencing with bioinformatics analysis, fact that enhances the reliability and depth of their research.The study effectively presents results using well-prepared tables and representations, which enhance data interpretation and readability.

THe study merits publication in its present form

Author Response

Dear Reviewer

Thank you very much for taking the time to review this manuscript.  Please find the corresponding revisions/corrections highlighted in yellow in the re-submitted files

Reviewer 5:

Comments: The study gives a new sight into the microbiota of Hirudo verbana, their feeding habits as well as medical uses. The identification of opportunistic pathogens (Rickettsia, Anaplasma, Treponema) in H. verbana contributes to new knowledge in microbiology. However, research on the leech microbiota is not entirely new, so the originality comes mainly from the comparative approach (wild leeches vs. farmed leeches) and pathogen identification. Their research underscores the requirement of systematic testing of leeches before medical application based on strict regulatory policies and clinical practices. The paper is well structured and written, with a logical flow from methodology to findings. Their methodology is robust by integrating high-throughput sequencing with bioinformatics analysis, fact that enhances the reliability and depth of their research. The study effectively presents results using well-prepared tables and representations, which enhance data interpretation and readability.

The study merits publication in its present form.

 Response:  We thank the reviewer for his constructive comments!

Round 2

Reviewer 3 Report

Comments and Suggestions for Authors

In Table 3, the Shannon index and its standard deviation should be rounded to hundredths, and the Simpson index and its standard deviation should be rounded to thousandths.
In Table 2, the image is blurry, the names are underlined with wavy lines.
The authors did not compare all samples with all in Figures 6 and 7, this is unacceptable.

Author Response

Comments to Reviewer 3 (Round 2):

Comments 1: In Table 3, the Shannon index and its standard deviation should be rounded to hundredths, and the Simpson index and its standard deviation should be rounded to thousandths.
Response 1:  We have revised the table accordingly: The Shannon index and its standard deviation are now rounded to the hundredths place. The Simpson index and its standard deviation are now rounded to the thousandths place.

Comments 2: In Table 2, the image is blurry, the names are underlined with wavy lines.
Response 2: The table 2 was  improved and corrected.

Comments 3:  The authors did not compare all samples with all in Figures 6 and 7, this is unacceptable.

Response 3 :

Figer 6: as per the reviewer’s request, we have conducted Kruskal-Wallis tests to compare all groups, including body regions, specifically for environmental bacteria. The statistical results indicate no significant differences (p > 0.05) in the abundances of environmental bacteria across these categories. Therefore, we conclude that the environmental bacterial composition remains consistent across 3 different groups.

Bacteria,H-stat,p-value

Comamonadaceae,3.4093567251461994,0.33270992576097974

Nubsella,1.0512820512820582,0.788845853065442

Rhizobiaceae,0.47938751472321467,0.92339597932005

Chitinophagaceae,1.0973236009732432,0.7777202159998123

As there was no significant variation in bacterial abundances among different body regions (p > 0.05),  comparison in Figure 6 is valid, as pooling body regions does not introduce bias.

Figer 7: 

As per the reviewer's request, we performed additional statistical analyses, including comparisons across body regions. The results indicate no statistically significant differences among body regions (p > 0.05). Therefore, we focused our analysis on 3 groups, which showed significant differences. The results in Figure 7 remain valid, with updated statistical values (H = 14.74, p = 0.00063).

Bacteria, Feeding Group, H-statistic,p-value

Mucinivorans, Farmed unfed,3.0,0.3916251762710877

Mucinivorans, Farmed fed,3.0,0.3916251762710877

Mucinivorans, Wild fed,3.0,0.3916251762710877

Aeromonas, Farmed unfed,3.0,0.3916251762710877

Aeromonas, Farmed fed,2.999999999999999,0.39162517627108917

Aeromonas, Wild fed,3.0,0.3916251762710877

Lactobacillales, Farmed unfed,3.0000000000000013,0.3916251762710884

Lactobacillales, Farmed fed,2.999999999999999,0.39162517627108917

Lactobacillales, Wild fed,3.0,0.3916251762710877

Vagococcus, Farmed unfed,2.999999999999999,0.39162517627108917

Vagococcus, Farmed fed,3.0000000000000013,0.3916251762710884

Vagococcus, Wild fed,3.0,0.3916251762710877

Morganella, Farmed unfed,3.0000000000000013,0.3916251762710884

Morganella, Farmed fed,3.0000000000000013,0.3916251762710884

Morganella, Wild fed,3.0,0.3916251762710877

Bacteria, Body Region, H-statistic, p-value

Mucinivorans, mouth,2.0,0.36787944117144245

Mucinivorans, pharynx,2.0,0.36787944117144245

Mucinivorans, crop,2.0,0.36787944117144245

Mucinivorans, intestine,2.0,0.36787944117144245

Aeromonas, mouth,2.0,0.36787944117144245

Aeromonas, pharynx,2.0,0.36787944117144245

Aeromonas, crop,2.0,0.36787944117144245

Aeromonas, intestine,2.0,0.36787944117144245

Lactobacillales, mouth,2.0,0.36787944117144245

Lactobacillales, pharynx,2.0,0.36787944117144245

Lactobacillales, crop,2.0,0.36787944117144245

Lactobacillales, intestine,2.0,0.36787944117144245

Vagococcus, mouth,2.0,0.36787944117144245

Vagococcus, pharynx,2.0,0.36787944117144245

Vagococcus, crop,2.0,0.36787944117144245

Vagococcus, intestine,2.0,0.36787944117144245

Morganella, mouth,2.0,0.36787944117144245

Morganella, pharynx,2.0,0.36787944117144245

Morganella, crop,2.0,0.36787944117144245

Morganella, intestine,2.0,0.36787944117144245
